# Analysis of HPV Integrations in Mexican Pre-Tumoral Cervical Lesions Reveal Centromere-Enriched Breakpoints and Abundant Unspecific HPV Regions

**DOI:** 10.3390/ijms22063242

**Published:** 2021-03-22

**Authors:** María Lourdes Garza-Rodríguez, Mariel Araceli Oyervides-Muñoz, Antonio Alí Pérez-Maya, Celia Nohemí Sánchez-Domínguez, Anais Berlanga-Garza, Mauro Antonio-Macedo, Lezmes Dionicio Valdés-Chapa, Diego Vidal-Torres, Oscar Vidal-Gutiérrez, Diana Cristina Pérez-Ibave, Víctor Treviño

**Affiliations:** 1Hospital Universitario “Dr. José Eleuterio González”, Centro Universitario Contra el Cáncer, Universidad Autónoma de Nuevo León, Av. Francisco I. Madero S/N, Mitras Centro, Nuevo León 64460, Mexico; maria.garzarg@uanl.edu.mx (M.L.G.-R.); dperezi@uanl.edu.mx (D.C.P.-I.); oscar.vidalgtz@uanl.edu.mx (O.V.-G.); 2Escuela de Ingeniería y Ciencias, Tecnologico de Monterrey, Ave. Eugenio Garza Sada 2501, Monterrey 64849, Mexico; mariel.oyervides@tec.mx; 3Departamento de Bioquímica y Medicina Molecular, Facultad de Medicina, Universidad Autónoma de Nuevo León, Av. Francisco I. Madero S/N, Mitras Centro Monterrey, Nuevo León 64460, Mexico; antonio.perezmy@uanl.edu.mx (A.A.P.-M.); celia.sanchezdm@uanl.edu.mx (C.N.S.-D.); 4Departamento de Ginecología y Obstetricia, Hospital Universitario “Dr. José Eleuterio González”, Universidad Autónoma de Nuevo León, Av. Francisco I. Madero S/N, Mitras Centro, Nuevo León 64460, Mexico; ana_bg88@hotmail.com (A.B.-G.); dr.mauro8207@gmail.com (M.A.-M.); dr.lezmes@gmail.com (L.D.V.-C.); diego.vidaltr@uanl.edu.mx (D.V.-T.); 5Escuela de Medicina y Ciencias de la Salud, Tecnologico de Monterrey, Av. Morones Prieto 3000, Colonia Los Doctores, Nuevo León 64710, Mexico

**Keywords:** cervical cancer, HPV integration, hot spots, cervical lesions

## Abstract

Human papillomavirus (HPV) DNA integration is a crucial event in cervical carcinogenesis. However, scarce studies have focused on studying HPV integration (HPVint) in early-stage cervical lesions. Using HPV capture followed by sequencing, we investigated HPVint in pre-tumor cervical lesions. Employing a novel pipeline, we analyzed reads containing direct evidence of the integration breakpoint. We observed multiple HPV infections in most of the samples (92%) with a median integration rate of 0.06% relative to HPV mapped reads corresponding to two or more sequence breakages. Unlike cancer studies, most integrations events were unique (supported by one read), consistent with the lack of clonal selection. Congruent to other studies, we found that breakpoints could occur, practically, in any part of the viral genome. We noted that L1 had a higher frequency of rupture integration (25%). Based on host genome integration frequencies, we found previously reported integration sites in cancer for genes like FHIT, CSMD1, and LRP1B and putatively many new ones such as those exemplified in CSMD3, ROBO2, and SETD3. Similar host integrations regions and genes were observed in diverse HPV types within many genes and even equivalent integration positions in different samples and HPV types. Interestingly, we noted an enrichment of integrations in most centromeres, suggesting a possible mechanism where HPV exploits this structural machinery to facilitate integration. Supported by previous findings, overall, our analysis provides novel information and insights about HPVint.

## 1. Introduction

Cervical cancer (CC) is the is the fourth most common malignancy and a leading cause of mortality in women worldwide [1]. In Mexico, CC is the second most common cancer in women [2]. High-risk (HR) human papilloma viruses (HPV) infection is the most important factor in developing CC [3]. HPVs belong to the *Papillomaviridae* family, and more than 225 types have been characterized based on sequence information [4]. HPVs are non-enveloped, circular, double-stranded DNA viruses of approximately 55 nm diameter [5]. HPV genomes are less than 8000 base pair (bp) with one non-coding, long regulatory control region and eight protein-coding genes: L1 and L2 encode capsid proteins, and E1, E2, E4, E5, E6, and E7 encode replication, transcription and transformation proteins [5,6,7].

HPVs are classified as low risk (LR) and HR, depending on the relative propensity to produce malignant progression [7]. LR-HPVs are associated with a spectrum of benign warts, whereas infections with HR-HPVs manifest by intraepithelial neoplasia that can produce malignant progression [7,8].

While most HR-HPV infections are cleared spontaneously by the immune system, a few will develop persistent infections that could progress to invasive disease [9]. Therefore, besides HR-HPV infection, other risk factors are required to initiate the transformation process [10,11]. The integration of HPV DNA into the host genome is considered a key risk factor for CC development [12,13].

HPV integrations (HPVint) are likely to occur through a microhomology-mediated DNA repair pathway, based on the evidence that the micro homologous sequence is significantly enriched around integrations sites [10]. Some reports have suggested that HPV promotes oncogenesis by disrupting tumor suppressor genes and chromosome instability [14,15]. It is thought that the HPV genome integrates into the host genome randomly at the beginning of HPV infection; however, in long-term infections, integrations are observed at recurrent loci [5,16]. These facts suggest that recurrent loci provide clonal selection advantages that contribute to carcinogenesis. The target host genes observed in these “hot-spots” are enriched in genes that are continuously expressed during transcription and DNA repair [14,17,18]. Several integrations sites are located inside the introns of tumor suppressor genes (like *SCAI* and *NR3C2*), likely altering the original expression patterns and contributing to the complete loss of gene function [15]. Several studies discover viral frequent integration sites in the *MYC*, *THEM49,* and *FANCC* genes [14]. However, other gene sites have been reported with a less recurrent frequency of integration [14]. Therefore, it is important to characterize the integration in all clinical stages to identify if these “hot-spots” are present in pre-cancerous lesions, at what proportion, and if they are associated with other clinical manifestations such as viral persistence.

The progression to CC by viral integration can also be affected by the HPV genotype. In invasive CC, the integration of HPV 31 and 33 was found less frequent than HPV 16, 18, and 45 [19]. Thus, HPV type had a strong effect on integration frequency, suggesting that the malignant potential is reflected by their integration frequency in invasive CC [19].

Because HPV is needed for tumorigenesis, understanding viral oncogenesis in pre-cancerous lesions is critical for clinical management and CC prevention. This study aimed to analyze HPVint in pre-tumor cervical lesions. HPVint status and locus have become important for discovering the mechanisms underlying virus infection, improving diagnosis, and cervical cancer treatment [20,21,22,23]. Moreover, the level of HPVint is positively correlated with cervical intraepithelial neoplasia grades and has been proposed as a marker for CC progression [10,24,25]. Therefore, HPVint status may be a biomarker for diagnosis, progression, and CC screening [17].

Briefly, in this project, we selected liquid cervical samples from colposcopy consultation patients and carried out high-throughput viral integration detection (HIVID), next-generation sequencing (NGS), and bioinformatics methods to identify viral integration sites, viral and human genome breakdown sites, and HPV types based on capturing HPV sequences using a set of viral-specific probes. In the HIVID method, the fragments carrying an HPV sequence are enriched by a set of HPV probes and then processed to high-throughput sequencing [26,27]. We followed a common cancer strategy [20,28] with specific customizations (Figure 1). We found multiple integrations and HPV types across samples.

## 2. Results

### 2.1. Patient Samples

A total of 24 cervical cell samples were taken from January 2014 to July 2016, DNA was extracted and stored at −20 °C until use. Samples were grouped according to Pap smear result: Normal, atypical squamous cells of undetermined significance (ASCUS), low-grade cervical lesions (LSIL), high-grade cervical lesions (HSIL), and Unknown (when Pap smear result was unavailable).

The patients included in this study were referred to the colposcopy consultation of the Department of Gynecology and Obstetrics of the Hospital Universitario “Dr. Jose Eleuterio González” of the Universidad Autónoma de Nuevo León (HU-UANL) due to an altered Papanicolaou test, genital warts, or genital pathology detection. The median age was 36 years (SD = 13) and had a mean body mass index (BMI) of 26.3, where half of these patients had overweight. The patients selected for this study had a sexual debut at a median age of 20 years old with an average of 3 sexual partners. Fifteen patients had a previous sexually transmitted disease reported in their clinical history.

### 2.2. HPV Genotyping by qPCR and Sequencing

HPV-positive samples were genotyped by multiplex qPCR. The method detects 14 HR-HPV types (HPV 16, 18, 31, 33, 35, 39, 45, 51, 52, 56, 58, 59, 66, and 68). To maximize detection, the reads were mapped to a de novo collection of HPV types (hpvDB) representing all know HPV sequences. The results are detailed in Appendix A.

We observed 58% HPV-type agreement between both detection methods (Appendix A). The HPV types most frequently detected by qPCR were HPV 39 and 52, whereas by HIVID-NGS were 51, 52, and a possible variant of 45 (t45).

### 2.3. HIVID-NGS HPV Genotype Detection

We noted that the 24 samples obtained more than 140,000 total reads (Table 1), 67% showing length of 142 nt (85% >= 120 nt) with a median insert size of 171 nt (median absolute deviation of 55 nt). With these overall sequencing results (140K reads, 120 nt each), we could reach depths around 2100 per nt for a typical HPV genome (8000 nt), or at least depths around 630 considering 30% mapping reads, which seems reasonable to estimate HPV presence. Thus, to determine possible HPV types per sample by our capture-sequencing strategy, we first mapped the reads to the hpvDB containing 451 representative HPV types. The results are shown in Table 1. On average, the mapped reads were 33%. A sample showed less than 500 total mapped reads (M-7449), suggesting that HPV typing is uncertain. The number of detected HPV types was not correlated to the number of reads per sample (R^2^ = 0.00, *p* = 0.81), mapped reads (R^2^ = 0.03, *p* = 0.43), or % of mapping (R^2^ = 0.00, *p* = 0.81), suggesting that the number of types detected is not highly dependent on sequencing reads. Most samples showed multiple HPV infections (92%), similar to 93% found in liquid-based cytology specimens using HPV capture technology [21]. We noted 15 HPV types in two or more samples (Table 2). To determine whether HPV type detection in several samples is not the result of cross-contamination, we first observed that the pattern of HPV types in samples was not a subset of another sample. For example, after analysis of the five samples for HPV 51 (M-7454, M-7457, M-7460, M-7462, M-7472), all showed different patterns of detected HPV types, indicating that there was no contamination between them (see Table 1). Moreover, we compared the particular nucleotide variant sites observed in HPV 51, 11, 52, and 16 (Figure 2). In all cases, each sample can be distinguished from the others by particular nucleotide variant sites. Thus, these results demonstrate that HPV typing by sequencing is not product cross-contamination. For t45 (MP134365 showing oligonucleotide frequencies similar to HPV45), we noted detections in 10 samples in more than 1% but at a low number of reads (Table 1 and Table 2). Only two regions of the genome were mapped (around 1500 and 7400). Still, reads do not show evidence of cross-contamination (Appendix A). The low number of reads and the high number of nucleotide variant sites suggest that a similar but distinctive type might be present instead of HPV45 or t45.

In samples with a high number of reads, we observed complete coverage of reads all along the HPV genomes (Figure 2). Interestingly, we noted a highly conserved pattern of depth across samples within HPV types. Because our results show that cross-contamination is unlikely, the similarity in the depth pattern is more likely due to biases of the capturing, sequencing, mapping, and database used rather than to the relative presence of HPV genomic regions.

### 2.4. Identification of Viral Integrations

In our preliminary analyses, we tried common approaches that we have already used in cancer to estimate integrations [20]. Nevertheless, we observed almost no reported integrations even though we manually observed that unmapped and partially mapped reads contained HPV and human sequences. We reasoned that current pipelines for viral integrations methods are focused on cancer samples [20,29,30,31] where cells have already passed through a heavy clonal selection process. Therefore, we did not follow these methods. Instead, we devised a novel approach considering only not fully mapped reads and whose one end matched to HPV and the opposite end matched to human (see Methods and Figure 1). Using this approach, we focus on reads that must contain evidence of an integration point within the read. Then, candidate reads were aligned (using *blastn*) to human and HPV to determine specific read fragments at nucleotide level corresponding to each genome. Finally, reads were filtered as described in methods to diminish false positives per sample and only possible integrations involving HPV types reported in Table 1.

Figure 3A shows examples of detected integrations in two known recurrent cancer genes, *RAD51B* [20] and *MACROD2* [32]. Figure 3B shows two reads of different lengths (no PCR duplicates) mapping to the same positions close to gene *RAB32*. Overall, the putative integrated reads across samples are summarized in Table 3. Relatively, the median percentage of integration was 0.06% of the mapped reads. The maximum was 1.71% observed in a carcinoma sample (M-7440), followed by 1.52% in a HSIL sample. The minimum was 0.02% observed in a LSIL sample (M-7445) followed by 0.03% in a PAP-normal sample (M-7471). We used the hits to chromosome Y (chrY) as an indicator and estimation of false-positive rate. The median of hits to chrY was 0.2% (0–0.7%), approximately to 1 per 500 hits, which is approximately 10 times less than expected by random chance (1.9% relative to the chrY size), suggesting that our pipeline is precise.

We observed that the percentage of integrated reads per type highly corresponded to the overall mapping shown in Table 1.

To determine whether there are preferences for integration sites in HPV, we compared the overall mapping to the putative integrated reads. We noted that peaks of putative integrated reads highly correspond to peaks of mapped reads across samples, HPV types, and viral load, as shown in representative comparisons in Figure 4 besides those shown in Figure 3. For example, the sample M-7447 shows clear peaks of integrations in HPV52 and HPV74 correlated to the overall sequencing maps’ peaks. Moreover, we observed similar behavior in low load types (HPV87 for M-7447 sample in Figure 4). This result suggests that integration at this stage of the disease has not been highly selective for specific HPV genome regions.

Different from cancer samples [19], we rarely observed repeated reads for a specific integration. This result is consistent with the idea that lesions still have not been through clonal selection. Nevertheless, in manual revision, we noted two different detected reads of a putative integration close to the gene *RAB32* (Figure 3B). One of these reads was marked by our pipeline containing 35% of their sequence within a detected tandem repeat region (TG in Figure 3B). Therefore, we systematically estimate the integration points showing more than one read of evidence after removing reads in repeat regions. We found 372 regions in 13 samples close to regions of 354 human genes (Appendix A). Only eight gene regions in 9 samples were found to have two or more reads in 2 or more samples (Figure 5A). Because some reads do not seem to be PCR duplicates (such as shown in Figure 3B), these results suggest that low numbers of cells have been divided after the integration.

Because some genes have shown a higher frequency of HPVint in cancer [10,20,32,33,34], we wonder whether, in lesions, integrations can also be detected in high-frequent genes. To improve certainty, we only considered coding genes showing two or more reads in the same sample independently of the exact integration point. We found 2,616 genes in 13 samples (Appendix A). To highlight the most frequent genes, we selected those genes present in 4 or more samples (Figure 5B). We noted previously reported and novel genes. For example, *FHIT*, *CSMD1*, and *LRP1B* (Figure 6) have been reported in cancer and lesions [10,35]. Nevertheless, Figure 5B shows more than 30 genes not previously reported found in 4 samples or more. Figure 6 shows examples of *CSMD3*, *ROBO2*, and *SETD3*. The integration point in *SETD3* seems to be the same across samples and HPV types. Nevertheless, the read sequences show very similar hits in a close and apparently duplicated region at ~10 kb distance.

We also counted integrations in non-coding regions that showed two reads or more per sample in regions around 10 kb From the 5172 regions in 18 samples (Appendix A), we highlighted those present in 3 or more samples (Figure 5C). Interestingly, we noted that 20 of the 28 regions shown are located very close to chromosomal centromeres. An overall map of integrations confirms that centromere regions are rich in integrations (Figure 7). This observation is consistent with previous reports linking HPV proteins with centromere and kinetochore [36,37].

From HPV types that were annotated, we identified that the most frequent viral integrations involve L1 region (25%), followed by L2 (17%), E7 (16%), E1 (14%), E2 (13%), E5 (6%), and E6 (5%). The genome region that had the least rupture percentage was E4 (4%).

## 3. Discussion

### 3.1. HPV Genotyping

CC remains a public health problem in Mexico since it is the second cause of death from cancer in women [2,38]. Persistent infection with HR-HPV produces pre-cancerous lesions starting with low-grade squamous intraepithelial lesions (LSIL), progressing to high-grade squamous intraepithelial lesions (HSIL) until CC is generated [39,40]. Although infection with HR-HPV is necessary to develop the oncogenic process, its genome integration is considered one of the most important risk factors for cervical carcinoma development [10]. Therefore, in this study, we analyzed the HPVint in pre-tumorous lesions in Mexican women. It is important to mention that when we compared observed HPV with nucleotide variant sites in each sample (for example those shown in Figure 2), we distinguished them from each other by presenting particular variant sites. Thus, the HPV typing by sequencing and the subsequent HPVint are not a product of cross-contamination.

HPV genotyping by NGS and qPCR were in agreement in about 58% of patients. qPCR is limited by the primer sequences, whereas NGS is limited by the probe sets. Because the probe set is comprehensive among types and across the genome [10], NGS should be more accurate to detect HPV types. Nevertheless, it is also limited by the database used to map reads. In this context, we preliminarily compared the reads mapped into the hpvDB used containing 451 HPV sequences against a database of 4342 HPV sequences (10x more sequences), resulting in only 6% improvement in mapped reads for 20 of the 24 samples, suggesting that the database used and the HPV typing were adequate. The use of large collections of sequences, such as all 4,342 sequences, generates the problem that many reads are assigned to very similar sequences fragmenting the estimation of correct types. Even in hpvDB of 451 sequences, we observed fractions of reads assigned to t52 rather than to HPV52, presumably due to minor sequence changes. By NGS, the most frequent HPV genotypes found were HPV 51, 52 plus t52, and HPV 11. We observed t45 (MP134365) in less than 1% on 11 samples or a low number of reads in other samples. Only two regions of ~300 nt were mapped. t45 is a sequence similar to HPV 45, which in turn is similar to HPV 18 and 59 [4]. We noted co-occurrence of t45, with HPV 18 and 59. Thus, it is likely that the t45 reads correspond to other genotypes putatively similar to HPV 18 and 59 or to a particular type common in our population. These subtle details need to be solved.

In our study, 13 of 24 patients had LSIL, and 8 had multiple HPV infection (61%). In a study with LSIL patients, the most prevalent HPV genotypes were 66 (25%), 16 (21%), 53 (18%), 51 (17%) and 52 (14%) [41], while a Mexican study showed 16 as the most common (26.3%), followed by 31 (11.5%), 51 (10.6%), and 53 (10.2%) [42]. In our patients, the most frequent HR-HPV genotypes in LSIL were HPV 39 (n = 6), HPV 52 (n = 4), and 56 (n = 4), the same results were obtained by both methods (NGS and qPCR).

Another interesting finding in this study was that most all the patients had multiple HPV infections by the NGS method (92%). Our results are in good agreement with a previous study using NGS for HPV genotyping that also found that NGS had high sensitivity, in particular for multiple infections [43,44]. Other authors found multiple HPV infections in 98% of the liquid-based cytology samples with precancerous lesions [21]. The HIVID-NGS method we used can detect multiple infections and unknown or incompletely characterized types, for which sequence data are not available.

### 3.2. HPVint

HPV is frequently integrated into the human genome [17]. The integration of high-risk human papillomavirus (HR-HPV) into the host genome is seen in ~85% of cervical squamous cell carcinomas. It is viewed as a critical driver of squamous carcinogenesis [45]. Therefore, the integration of HPV is considered an event that promotes cellular carcinogenesis [12,22,46]. Most of the HPVint studies are focused on carcinomas [20,29,30,31] where clonal selection for tumor growth has already happened, and an integration event showed several non-PCR duplicates reads. In this study, nevertheless, we analyzed the viral integration in precancerous samples [16,21,47] where a new analysis strategy was designed that allows us to identify reads with evidence of the viral integration with presumably high sensitivity (Figure 1). It was found that virtually all samples analyzed presented viral integration (96%) (except one sample of scarce mapped reads). Moreover, several of the samples (63%) showed integrations of two or more viral types. Our data reveal that HPVint occurred in early stages before carcinogenesis, which agrees with other reports, where HPVint was observed in early stages and that the rate and number of integrations increases according to the progression of the disease [10,13,24,48]. We included three HPV positive samples with normal Pap, all of them showed integrations. Although we found a higher proportion of integration in the early stages than has been reported, this may be due to differences in the sample characteristics, the sensitivity of the assay method, the database, and the pipeline used. NGS combined with capture technology seems to be the most sensitive method to detect HPV viral integrations [21,49,50,51,52]. It would be interesting to study whether specific integrations are related to HPV infection persistence or other clinical characteristics.

### 3.3. Centromere Integrations

We noted an enrichment of integrations in most centromeres (Figure 6 and Figure 7). There is evidence linking HPV proteins to the centromere and kinetochore [36,37]. This fact suggests a possible mechanism where HPV utilizes some of the structural machinery to facilitate its integration. In this regard, the data obtained in our study seems to be scarce to answer further questions. Nonetheless, a specific experimental design where deeper sequencing is performed in wisely selected samples may be adequate. In our data, we noted an enrichment in most centromere chromosomes except in chr3, chr5, and chr13, and low in chr12 and chr20. It would be exciting whether this may be related to particular HPV types, stages of the disease, populations, chromosomes, or technical issues. In our data, the enrichment seems to be around 3 to 30 times higher in centromeres than in other parts of the human genome.

### 3.4. HPVint Ratio and Breakpoint

Other authors noted that the integrations’ frequency increases with lesion progression [16,53,54]. Thus, an important issue in the analysis of HPVint is the fraction that may be present in integrated form compared to that in episomal. A recent analysis used the fraction of reads with integrations relative to those showing no integration in the same region as the episomal [21]. This estimation seems correct, assuming that many cells carry the same integration event, and the integration site is unique. These assumptions are acceptable for cancer or tumors where clonal selection has driven cellular expansions. However, for non-tumoral cells, applying the same criteria could, presumably, result in highly deviated estimations. In this context, we interestingly observed that depth of integrated reads along the HPV genomes correlates with overall depth. This observation suggests that choosing an integration point in the HPV genome is close to a uniform random process, which agrees with other studies [10]. Consequently, the expected fraction of integration reads in about 8 kb length is 0.0125% per breakage point (nucleotide). One breakage point would open its circular episomal form to generate a linear molecule that can be integrated into the host genome. Two or more breakage points would generate smaller fragments. We observed that the minimum integration read fraction was approximately 0.02% and 0.03% for an LSIL and Normal pap smear patients, respectively, which corresponds to about 2 breakage points. On the other hand, we also observed samples with 10 times more integrated read fractions of 0.12% for an LSIL patient and even 100 times more than one breakage point to about 1.52% for an HSIL patient. We noted that the two HSIL samples showed high integration fractions (0.08% and 1.52%), which can be achieved by raising the HPVint activity or by an infection that persists over time, aggregating integrations. In any case, the integration fraction could be used as a surrogate measure for future experiments or studies.

Other study in liquid-based cytology specimens found 10 genotypes (HPV30, 35, 39, 44, 45, 53, 56, 59, 74 and 82) that were detected only in episomal form [21]. They also include HSIL, ASCH, LSIL, ASCUS, and cancer samples. We found HPVint in all these genotypes.

### 3.5. Comparison of Reported HPV Genes vs. Observed

Consensus reports reached that various portions of the HPV genome are deleted in the integrated HPV sequences [55]. Common disruption of the viral E2 gene has been demonstrated in different studies, resulting in functional inactivation. Loss of the E2 expression abrogates the E2-mediated repression of E6/E7 transcription from integrated HPV DNA and increases the expression of these oncoproteins that induces HPV-immortalization [3,55,56]. Globally, we found that breakpoint could occur in any part of the viral genome, which agrees with what was previously reported with the same methodology we used [10], with a higher frequency in the L1 region (25%), followed by L2 (17%), E7 (16%), and E1 (14%), and less frequency in E6 (4.6%), and E4 (4%). Other researchers developed the HIVID-NGS methodology that we used [10], finding a higher frequency of rupture in E1. Another study in cervical lesions found that the disruptions were more frequent in the L2 gene (67.7%), followed by the L1 gene (25.8%) and the E1 gene (22.6%) [16]. When we analyze the samples individually, we observe differences in integrations. In some samples, we observe a higher disruption in E2, L1, or L2. However, we also found samples in which E2 is not broken. The differences compared with our results may be due to the histological classification of the samples, viral genotype, database, and pipeline used. Our view is that there is variability between patients, viral strains, and other unknown factors such as lifestyle.

### 3.6. Comparison of Reported Human Genes

Viral genome integration into the host genome triggers various genetic alterations, such as oncogenes amplification, tumor suppressor gene inactivation, and inter- or intra- chromosomal rearrangements, as well as genetic instability [14,33]. The integration event could lead to uncontrolled growth, which can eventually lead to cancer [15].

It has been suggested that HPV initially integrates into the human genome randomly, based on its accessibility to the genome [10,13]. However, cancer studies have shown that the integration of the viral genome into some loci or hot spots is recurrent and can confer a selective growth advantage [10,33].

In our study, we found that the integration of the viral genome was almost non-selective when analyzed by large segments, finding breakpoints throughout the entire human genome. However, when analyzed by gene, we also detected similar integrations in various samples. It was found that in 2,616 coding genes, 2 or more integration sites presented among samples (Appendix A), like *RAD51B*, *MACROD2*, *FHIT*, *CSMD1*, *LRP1B,* and *DLG2* (Figure 5 and Figure 6), which had already been previously reported as frequent sites of integration in squamous carcinoma samples [10,33,34]. The *RAD51B* gene is implicated in the DNA repair pathway; viral integrations in this gene cause loss of function and have been associated with other types of cancer, like breast, ovary, prostate, and colorectal [14]. Other authors also found integration in *FHIT* and *LRP1B* [10]. Intronic HPV breakpoints in *FHIT* and *LRP1B* have been related to decreased protein expression in carcinoma samples [8]. Also, it has been reported that HPVint in *MACROD2* may cause gene loss of function and impact genome instability [33]. Non-coding and structural variations in the *MACROD2* gene have been associated with cancer predisposition, especially colorectal cancer, reported to alter DNA repair [33,57]. Thus, our results are consistent with previous findings indicating that our data showing novel results are valuable.

Nevertheless, we found viral integration in more than 30 genes not previously reported, like *CSMD3*, *ROBO2,* and *SETD3* (Figure 5B). *CSMD3* is involved in dendrite development [58] and germline variants in this gene have been linked to colorectal cancer [59]. *ROBO2* participates as axon guidance, and cell migration [60] and low levels of mRNA expression are associated with poor survival in pancreatic ductal adenocarcinoma [61]. Another interesting finding was that the integration point in *SETD3* was the same for all samples and HPV type, which could be a pattern of integration in precancerous samples. An increase in the number of samples is required to corroborate this finding. This gene has been identified as an actin-specific histidine N-methyltransferase, and its expression has been associated with oncogenesis, especially in breast cancer [62]. Even though these three genes are involved in carcinogenesis, the meaningful impact in cervical cancer is not clear.

The interpretation of the integrations for many other genes and regions observed in various samples could be limited for various reasons. First, we used reads that show at least 30 not-mapped nucleotides. Although we observed no apparent bias in mapping, we noted some reads that could match similar regions of the genome. In this context, larger reads may be helpful. Second, we only used reads having evidence of the integration because we would like to quantify specific positions while paired reads mapping to different genomes are not accurate and to provide confidence in the use of a novel pipeline in non-tumoral samples. Nevertheless, paired reads may also be informative. Third, there is a lack of models to estimate the frequency of integrations expected by chance correcting for gene size, perhaps sequence context, and sample reads (indeed, we observed a high variation in the number of integrations among samples, from a handful to thousands).

## 4. Materials and Methods

### 4.1. Study Population

We selected 24 patients with gynecological alterations referred to the colposcopy consultation at the Gynecology and Obstetrics Department HU-UANL in Monterrey, Nuevo León, México. All of them agreed to participate in this study by signing an informed consent, previously approved on August 10, 2011 by the Institutional Review Board of the Hospital Universitario “Dr. Jose Eleuterio González” of the Universidad Autónoma de Nuevo León (Project identification code BI11-002). The PAP smear result was re-interpreted from transcripts in all but two samples. All the patients included were previously detected as HPV-positive [63].

### 4.2. Sample Collection and DNA Extraction

Cervical samples were taken using a cytobrush (Colpoltre^®^), preserved in ThinPrep^®^ PreservCyt solution, and stored at −70 °C until DNA extraction. Samples were collected from January 2014 to July 2016. DNA was extracted from cervical cells using the PureLink Genomic DNA kit from Invitrogen^®^ (Life Technologies, Carlsbad, CA, USA), following the manufacturer’s instructions. DNA quantity and quality were measured by spectrophotometry. DNA was stored at −20 °C until use. Samples were grouped according to Pap smear as Normal, ASCUS, LSIL, HSIL, and Unknown.

### 4.3. HPV Genotyping

HPV detection was performed using the consensus primers set PGMY 09/11, which amplifies a broad spectrum of HPV types. *β-globin* gene was used as an internal control. PCR products were analyzed by 2% agarose gel electrophoresis, stained with ethidium bromide, and visualized in a UVP Model 2UV High-Performance Transilluminator (Upland, CA, USA).

qPCR was performed to genotype and quantify HPV types in samples, using AmpliSens^®^ HPV HRC genotype-titer-FRT kit (Ecoli, Bratislava, Slovak Republic) according to the manufacturer’s instructions in a 7500 Fast Real-Time PCR System (Thermo Scientific, Waltham, MA, USA).

This kit is based on simultaneous real-time amplification (multiplex PCR) of DNA fragments of 14 HR-HPV types (HPV 16, 18, 31, 33, 35, 39, 45, 51, 52, 56, 58, 59, 66, and 68) and a DNA fragment of the *β-globin* gene as an internal endogenous control, carried out in four separated reaction tubes. Each genotype is detected in a separate fluorescent channel (FAM, JOE, ROX, and Cy5) that allows its detection and quantification (a total of 16 qPCR probes are measured). Each genotype is detected in a separate fluorescent channel, making it possible to determine the genotype and viral load.

### 4.4. Capturing and Sequencing

HIVID-NGS was performed by MyGenostics (Beijing, China) Gene Technology Company, as reported previously [10]. A target HPV gene region capture, and sequencing were performed using the ViralCap_HPV kit for capturing HPV genomes of 32 types (6, 11, 16, 18, 31, 33, 35, 39, 45, 52, 56, 58, 59, 66, 68, 69, 82, and others) developed by MyGenostics (Beijing, China) Gene HIVID-GNS method, HPV-specific probes are used to capture virus and flanking sequences prior to unbiased PCR amplification and NGS. It produces high-quality genotype data. Increased chance of finding HPVint due to sequence capture [26]. Technology Co., Ltd. This kit allows capturing the viral sequences in simultaneous detection of all known virus subtypes and virus variants, as well as information on the integration of the viral genome into the host genome. The libraries were constructed following the manufacturer’s instructions. DNA samples were sheared to 150–200 bp DNA fragments using a Covaris S2 system ultrasonicator (Covaris, Inc., Woburn, MA, USA). The fragments were purified, end blunted, A-tailed and adaptor-ligated. Libraries were hybridized with HPV probes, including 32 types of HPV, and then washed to remove uncaptured fragments. The eluted fragments were amplified by PCR to generate libraries for sequencing. Libraries were quantified then sequenced in the Illumina NextSeq500 high-throughput sequencer according to the manufacturer’s instructions (Illumina Inc., San Diego, CA, USA). The length of 91% of reads was 142. The insert size median was 171, and the median absolute deviation of 55.

### 4.5. HPV Mapping and Typing from Sequencing

A set of most dissimilar HPV types genomes was generated from Papillomavirus Episeme (PaVE) [64], a curated viral database dedicated to HPV, and from the NCBI nucleotide database. A clustering approach from oligonucleotide frequencies was used to generate a database of 451 representative HPV genomes (hpvDB). The database is included in Appendix A. The reads were mapped to hpvDB using *bwa-mem*. Overall estimations of viral presence were estimated mapping raw reads to the hpvDB. The most abundant HPV type was always considered. To include other types, we integrated those whose counts were at least 1% of the most abundant type and higher than 50 reads or that the read count was at least 5000.

### 4.6. Viral Integration Pipeline

Viral integrations pipeline methods are focused on cancer, which are based on multiple reads supporting the same integration point [20,29,30,31]. Therefore, we designed a specific data analysis pipeline for pre-malignant cervical lesions for viral integration in capture-based sequencing. Here, we will briefly describe the pipeline schematized in Figure 1. First, because our methodology is based on capturing HPV sequences, we focused on reads not fully mapped to hpvDB, after removing adaptor sequences. This process was performed by analyzing the CIGAR field of SAM/BAM files (https://github.com/samtools/hts-specs, accessed on 4 January 2021) taking reads with 30 nt or more not matched. Second, we generated pseudo-reads of 30 nt from 5′ and 3′ ends from NGS reads that did not fully map to hpvDB. Third, we identified *candidate integration reads* whose one end mapped to human (hg19) and the other end mapped to hpvDB. Fourth, original NGS candidate reads were aligned to human and hpvDB using *Blastn* [65]. The procedure was performed independently on each of the two paired reads (Figure 1). Fifth, we then filtered those blast hits showing a misalignment size less than 20, showing mismatch ratio less than or equal to 1/15, showing duplicate start-end positions in the same target sequence (taking the most significant), or, in HPV, showing alternative alignments for same target sequence (taking the most significant). This filtering was independently performed to human and HPV blast hit lists and finally removing those reads not found in both. In case of multiple hits per read into either human or hpvDB, we analyzed the top 3 most significant hits for annotation. To determine the most sensible hit, we used the annotation of the paired reads to resolve the ambiguous hit within the 3 most significant when possible, or used the most significant from blast otherwise. Sixth, reads were marked when they have more than 33% of their sequence in repetitive regions, when involving chromosome Y, or where both paired reads show integration, and the direction of mapping does not correspond to the expected inverted direction. To determine repetitive regions, we used Phobos setting parameters for detecting regions of size 12 nt or more (--searchMode imperfect --minLength 12 --minScore 2 --minLength_b 2 --minScore_b 2 --recursion 5 --outputFormat 3 --printRepeatSeqMode 2 --reportUnit 0 --mismatchScore −3 --indelScore −4).

## 5. Conclusions

In this study, our results had revealed characteristics of HPVint in precancerous lesions. We observed that breakpoint in HPV can occur in any part of the viral genome, with a higher frequency in L1 gene. We rarely observed repeated reads for a specific integration site, contrary to what is observed in cancer samples. This observation is consistent with the idea that lesions still have not been through clonal selection. Based on host genome integration frequencies, we found previously reported integration sites like *FHIT*, *CSMD1,* and *LRP1B* and others such as *CSMD3*, *ROBO2,* and *SETD3*. We noted an enrichment of integrations in most centromeres, suggesting a possible mechanism where HPV utilizes centrosome or kinetochore machinery to facilitate integration. We observed that in precancerous cervical cells, there are already integrations in genes observed after clonal selection. The integration sites identified in the host genome could be used as possible biomarkers for early diagnosis in patients with cervical lesions.

This study provides a theoretical basis to understanding the mechanism of tumorigenesis from the perspective of HPVint and its association with cervical lesions.

## Figures and Tables

**Figure 1 ijms-22-03242-f001:**
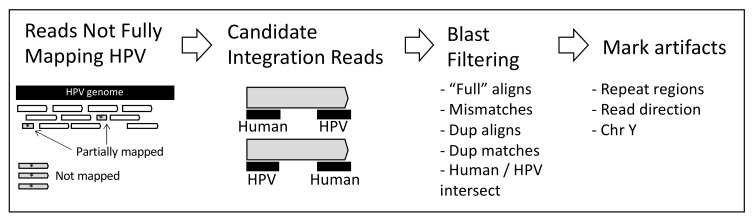
Scheme of the strategy used to identify reads showing evidence of HPVint.

**Figure 2 ijms-22-03242-f002:**
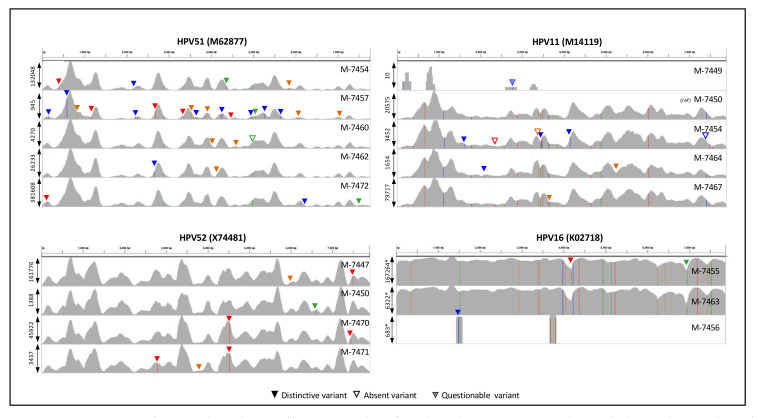
Comparison of mapped reads in different samples of 4 selected HPV types. Each panel shows the number of mapped reads (in vertical) of 5, 5, 4, or 3 samples detected in corresponding HPV types (NCBI accession number is shown). The vertical axis marked with “*” in HPV16 is shown in logarithm scale to clarify marked nucleotide variant sites. The peak number of reads per sample is shown at the left. The sample is shown at the right. Vertical lines in colors indicate sequence differences. Triangle marks show distinctive or absent variant sites.

**Figure 3 ijms-22-03242-f003:**
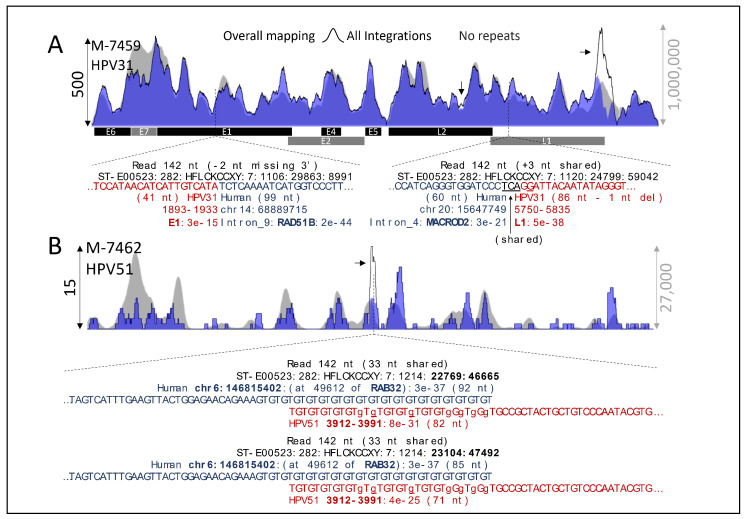
Examples of HPVint. (**A**) Two integrations at different known cancer genes (*RAD51B* and *MACROD2*). (**B**) Two detected integrations at the same position of the *RAB32* gene of reads showing different lengths. Note the region rich in GT. Arrows denote marked reads within detected tandem repeats regions. Nucleotides in lowercase denote mismatches relative to the target sequence; if underlined, refer to insertion relative to the target. The vertical axis at the left refers to integrations, whereas the right axis refers to overall HPV mapping.

**Figure 4 ijms-22-03242-f004:**
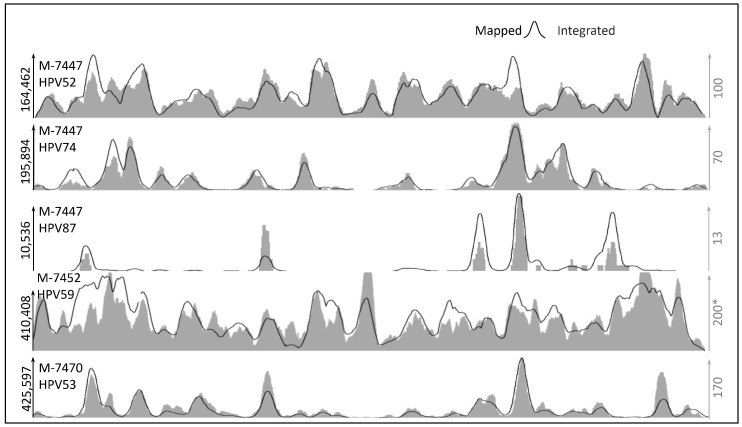
Comparison of mapped and putatively integrated reads. The vertical axis shows the number of reads. The horizontal axis represents the HPV genome coordinates from 0 to 8000 bp approximately. * Scale for M-7452 in integrated reads is reduced to half to clarify correlation (3 peaks are cut).

**Figure 5 ijms-22-03242-f005:**
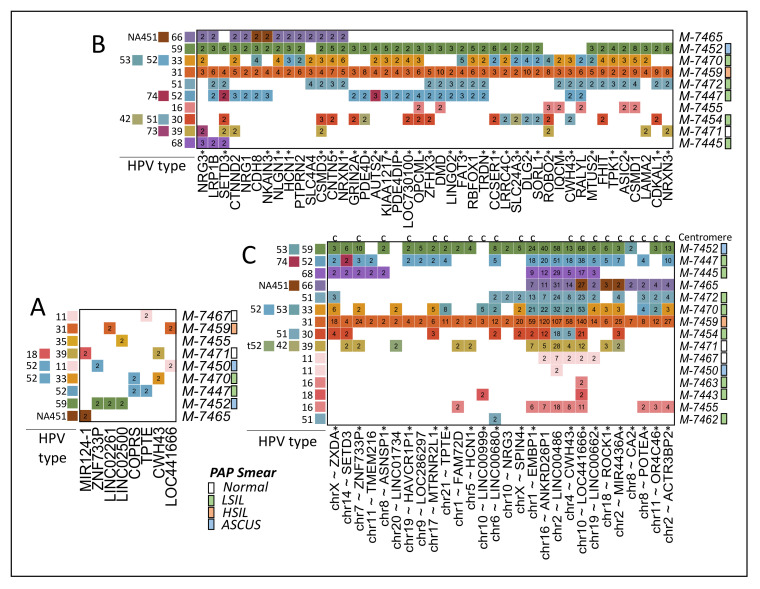
Multiple reads or integrations. (**A**) Reads in genes showing more than 1 read in the same integration point in 2 or more samples (targets in human region and HPV are the same). (**B**) Reads or integrations in coding genes found in 4 or more samples where HPV target type or precise human integration point may differ. Gene region includes 5 kb in 5′ and 3′ of the canonical transcript. Genes marked with a star “*” do not show records of co-occurrence with HPV in abstracts from PubMed. (**C**) Reads or integrations in non-coding regions close to genes in 3 or more samples where HPV target type or precise human integration point may differ. Regions marked with “*” and “c” are close to centromeres. Pap smear results per samples are shown.

**Figure 6 ijms-22-03242-f006:**
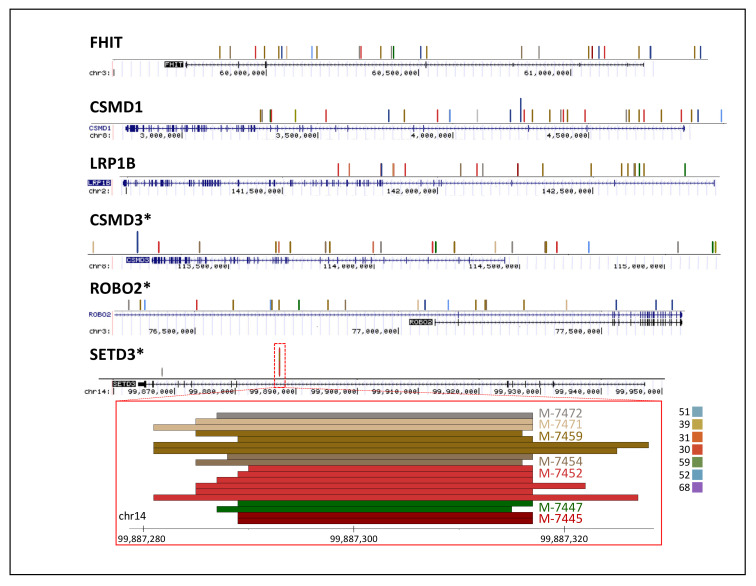
Examples of multiple integrations in known and novel genes. In each gene, a colored vertical segment represents a read. Different segment colors represent different samples. A specific region in the *SETD3* gene is detailed. Samples and corresponding integrations for HPV types are Scheme 5.

**Figure 7 ijms-22-03242-f007:**
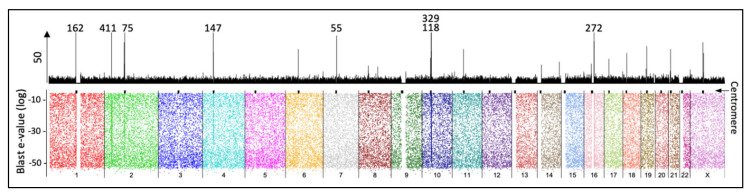
Overall map of 50,732 pooled read-integrations in the human genome. The top panel Scheme 20. bins). The vertical axis is cut to 50 to highlight differences. The numbers mark positions with higher than 50 counts. The bottom panel shows the integrations (dots) in all chromosomes distributed by blast e-value to show the lack of mapping bias. Centromeres coordinates (hg19) are marked with a black spot above. Note higher counts close to centromeres.

**Table 1 ijms-22-03242-t001:** Detected human papillomavirus (HPV types per sample.

Sample	Smear	Reads	Mapped *	%Map ^¶^	HPV Types ^+^	Percent ^+^
M-7440	CC	4,562,484	6474	0.1	t45	100
M-7443	LSIL	716,396	381,893	53.3	18, 74	99, 1
M-7445	LSIL	15,287,436	8,496,237	55.6	68, NA440, NA448, t45	42, 36, 22, <1
M-7447	LSIL	13,480,394	8,317,461	61.7	52, t52, 74, 87, 44, t45, t66	39, 38, 22, 1, <1, <1, <1
M-7448	ASCUS	417,768	29,188	7.0	6, t45	98, 2
M-7449	Normal	147,160	411	0.3	t45	100
M-7450	ASCUS	1,143,332	366,366	32.0	11, 52, t52, t56, 56, NA450	79, 8, 7, 2, 2, 1
M-7452	ASCUS	17,740,624	11,991,175	67.6	59, 89, 53, t45	98, 1, 1, <1
M-7454	LSIL	5,251,310	3,219,573	61.3	30, 51, 42, 11, t45	53, 38, 7, 2, <1
M-7455	NA	6,350,088	2,897,763	45.6	16, 35, NA446, t45	85, 8, 7, <1
M-7456	HSIL	3,879,422	8527	0.2	t45, 16	86, 14
M-7457	LSIL	367,094	13,371	3.6	6, t45, 51, 90	47, 44, 6, 3
M-7458	LSIL	272,030	18,674	6.9	6, 86, t45	92, 5, 3
M-7459	HSIL	28,572,654	21,901,105	76.7	31, t45	100, <1
M-7460	LSIL	494,080	56,676	11.5	51, 70, t45, 34	74, 22, 2, 2
M-7461	LSIL	365,174	2367	0.6	44, t45, 54, 53	49, 27, 13, 11
M-7462	LSIL	884,470	238,060	26.9	51, 54	96, 4
M-7463	LSIL	702,672	121,121	17.2	16, NA446, t45, 40	90, 8, 1, 1
M-7464	LSIL	481,510	40,073	8.3	11, 31, 34, t45	61, 30, 5, 3
M-7465	NA	9,826,324	6,183,616	62.9	t66, NA451, 66, t45	49, 31, 19, <1
M-7467	Normal	3,173,490	1,445,628	45.6	11, t45	100, <1
M-7470	LSIL	13,217,166	7,266,920	55.0	53, 33, t33, 52, t52, NA436, t45	39, 16, 13, 11, 11, 10, <1
M-7471	Normal	9,791,996	5,128,463	52.4	39, 82, t39, NA447, 73, 42, 66, NA449, 52, t52, t45	33, 27, 16, 10, 4, 3, 2, 2, 1, 1, <1
M-7472	LSIL	6,241,940	3,953,761	63.3	51, t45	100, <1

* Reads matching 85% or more. ^¶^ %Map: Percentage of total reads that were mapped. ^+^ HPV types sorted by corresponding percentage of reads. Percent show the corresponding fraction of reads identified in each HPV type.

**Table 2 ijms-22-03242-t002:** Detected HPV types in two or more samples.

Type	Samples	Reads	Within Sample % ^¶^
t45	M-7440, M-7445, M-7447, M-7448, M-7449, M-7452, M-7454, M-7455, M-7456, M-7457, M-7458, M-7459, M-7460, M-7461, M-7463, M-7464, M-7465, M-7467, M-7470, M-7471, M-7472	177,864	100, <1, <1, 2, 100, <1, <1, <1, 86, 6, 3, <1, 2, 27, 1, 3, <1, <1, <1, <1, <1
51	M-7454, M-7457, M-7460, M-7462, M-7472	5,433,163	38, 44, 74, 96, 100
52	M-7447, M-7450, M-7470, M-7471	4,197,311	39, 8, 11, 1
t52	M-7447, M-7450, M-7470, M-7471	4,018,079	38, 7, 11, 1
11	M-7450, M-7454, M-7464, M-7467	1,815,076	79, 2, 61, 100
53	M-7452, M-7461, M-7470	2,901,407	1, 11, 39
16	M-7455, M-7456, M-7463	2,561,367	85, 14, 90
6	M-7448, M-7457, M-7458	45,980	98, 3, 92
31	M-7459, M-7464	21,889,052	100, 30
t66	M-7447, M-7465	3,058,716	<1, 49
74	M-7443, M-7447	1,795,576	1, 22
66	M-7465, M-7471	1,298,463	19, 2
42	M-7454, M-7471	377,645	7, 3
NA446	M-7455, M-7463	219,032	7, 8
44	M-7447, M-7461	29,574	<1, 49
54	M-7461, M-7462	9174	13, 4
34	M-7460, M-7464	2977	2, 5

^¶^ Percent show the corresponding fraction of reads identified in corresponding samples.

**Table 3 ijms-22-03242-t003:** Detected integrations per sample and HPV type.

Sample	% Integration	Integrations	After Marks *	% Chr Y	Types ^¶^	Percents ^¶^
M-7440	1.71%	111	39	0.60%	t45	100
M-7443	0.04%	164	161	0.04%	18	100
M-7445	0.02%	1508	1491	0.02%	68, NA440, t45, NA448	96, 3.5, <1, <1
M-7447	0.05%	4554	4507	0.05%	52, 74, t52, 87, 44, t45, t66	77, 19, 3, <1, <1, <1, <1
M-7448	0.10%	28	26	0.09%	6	100
M-7449	0.73%	3	2	0.49%	t45	100
M-7450	0.04%	154	148	0.04%	11, 52, 56, t52, NA450	70, 21, 7, <1, <1
M-7452	0.07%	8249	7803	0.07%	59, 53, 89, t45	99, <1, <1, <1
M-7454	0.07%	2337	2292	0.07%	30, 51, 42, 11, t45	54, 39, 6, 1, <1
M-7455	0.06%	1790	1769	0.06%	16, 35, NA446, t45	91, 7, 2, <1
M-7456	1.52%	130	44	0.52%	t45	100
M-7457	0.04%	5	3	0.02%	90, 51	67, 33
M-7458	0.06%	11	7	0.04%	6, t45, 86	71, 14, 14
M-7459	0.08%	18,080	17,476	0.08%	31, t45	100, <1
M-7460	0.07%	38	34	0.06%	51, 70	74, 26
M-7461	0.17%	4	0	0.00%	-	-
M-7462	0.06%	137	129	0.05%	51, 54	98, 2
M-7463	0.10%	118	115	0.09%	16, 40, NA446	92, 6, 2
M-7464	0.05%	20	18	0.04%	11, 31	56, 34
M-7465	0.04%	2573	2539	0.04%	66, NA451, t66, t45	53, 36, 11, <1
M-7467	0.05%	737	691	0.05%	11, t45	99, <1
M-7470	0.08%	5952	5805	0.08%	33, 53, 52, NA436, t52, t45, t33	39, 39, 20, 1, <1, <1, <1
M-7471	0.03%	1754	1695	0.03%	39, 82, 73, 66, 42, 52, NA447, t39, t52, t45, NA449	67, 11, 10, 3, 3, 3, 1, <1, <1, <1, <1
M-7472	0.12%	4647	4418	0.11%	51, t45	100, <1

^¶^ HPV types sorted by corresponding percentage of reads. Percent show the corresponding fraction of reads identified in each HPV type.

## Data Availability

Some data results are provided in Appendix A.

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
