# Peer review of "Analysis of HPV Integrations in Mexican Pre-Tumoral Cervical Lesions Reveal Centromere-Enriched Breakpoints and Abundant Unspecific HPV Regions"

_ijms, 2021, doi:10.3390/ijms22063242_

Round 1
Reviewer 1 Report
The work deals with the important problem of HPV integration into the host genome in precancerous lesions. The authors used the Illumina NGS sequencing technique and a novel approach to compiling data from readings - using all known and deposited HPV genome sequences. Work is more cognitive than clinical or diagnostic. The introduction is very synthetically written, so the language seems rather clumsy.
Some comments on the results:
- verse 143-144 - the meaning of this sentence is difficult to understand. I understand that the pattern of the HPV types was different in each of the five samples mentioned, indicating that there was no contamination between them. - this should be corrected
- In Table 1 - sample M-7457 contains HPV 52 but the text and table 2 show that it should be HPV51 ??- this should be corrected
- In Table 2 - HPV11 type: % of reading inside M-7454 sample should be 53% (as shown in Table 1 but is 2%?? - this should be clarified
- Overall, all tables should be strengthened as they now appear to be part of the working protocol. The column names should be clearer. For example, in Table 1: What does "%Map, Percent, Types" mean?
- Figure 2 is to provide evidence that HPV typing by sequencing is free from contamination with other samples. The triangles representing the different HPV 51 variants are very faint - this needs to be corrected
The discussion is an important part of this work and has been well written.
Author Response
Reviewer 1
Comments and Suggestions for Authors
The work deals with the important problem of HPV integration into the host genome in precancerous lesions. The authors used the Illumina NGS sequencing technique and a novel approach to compiling data from readings - using all known and deposited HPV genome sequences. Work is more cognitive than clinical or diagnostic. The introduction is very synthetically written, so the language seems rather clumsy.
Some comments on the results:
- Verse 143-144 - the meaning of this sentence is difficult to understand. I understand that the pattern of the HPV types was different in each of the five samples mentioned, indicating that there was no contamination between them. - this should be corrected
Answer: We made changes to the wording of the paragraph (lines 141-143) to clarify that the HPV patterns were different in the five samples where HPV 51 was detected.
The wording was as shown below:
“For example, after analysis of the five samples for HPV 51 (M-7454, M-7457, M-7460, M-7462, M-7472), all showed different patterns of detected HPV types, indicating that there was no contamination between them (see Table 1)”
- In Table 1 - sample M-7457 contains HPV 52 but the text and table 2 show that it should be HPV51 ??- this should be corrected
Answer: Effectively, this was an error. Sample M-7457 was HPV 51. Thanks for finding this typo.
- In Table 2 - HPV11 type: % of reading inside M-7454 sample should be 53% (as shown in Table 1 but is 2%?? - this should be clarified
Answer: There was an error in the order of HPV types in Table 1, which was edited just before submission where the HPV types in the column were wrongly sorted by type number without re-ordering corresponding percentages. This was corrected and added a table footnote to clearly indicate that columns were ordered by %. Now, HPV11 in sample M-7454 correctly corresponds to 2% as shown in Table 2.
- Overall, all tables should be strengthened as they now appear to be part of the working protocol. The column names should be clearer. For example, in Table 1: What does "%Map, Percent, Types" mean?
Answer: We added table footnotes in Table 1 explaining that %Map indicate the percentage of total reads that were mapped, while Percent show the corresponding fraction of reads identified in each HPV type. Types were changed to HPV Types. Thanks for helping in clarifying.
The footnote of Table 1 is now as shown below:
* Reads matching 85% or more. ¶ %Map: Percentage of total reads that were mapped.
+ HPV types sorted by corresponding percentage of reads. Percent show the corresponding fraction of reads identified in each HPV type.
- Figure 2 is to provide evidence that HPV typing by sequencing is free from contamination with other samples. The triangles representing the different HPV 51 variants are very faint - this needs to be corrected
Answer: We increased the size of the triangles to improve the interpretation of the figure. We will revise the figure in proofreading to make sure the final resolution is sufficient.
- The discussion is an important part of this work and has been well written
Answer: Thank you for the comment.
Reviewer 2 Report
The manuscript presented by Garza-Rodriguez et al., describes the potential HPV integration sites in different cervical lesions. Certainly, the authors provide new insights for understanding the biological manners that could be potentially associated to cancer development. The manuscript is well written, and the methodology is very innovative, providing new approaches for expanding our knowledge in the field. Nevertheless, several important aspects are missing, and the manuscript must be improved before to be considered for publication.
Major concerns
- Complete information of tables 1 and 2 could go to supplementary information, these only show HPV detected in the studies population. Instead, authors could present this information as a new table indicating the frequencies of the HPV in the included analyzed cases to increase readability.
- Authors indicate that they distinguished variants in their samples, but they do not present such information, this information can be presented in a table due to the available information showing that variants confer a different prognosis in CC.
- Why the authors included the case with an unknown result of PAP, since they pretend to associate the integration with the different lesions. This must be clarified.
- It is not clear whether the integrations spots found in the study are associated to the different clinical stages analyzed in the study. Thus, authors need to present such information in an additional figure, since this is basically the main goal of this work.
- Authors indicate that these results could be associated to clinical stage, viral persistence and other clinical manifestations (lines 76-79) but is not covered in the manuscript. This needs clarification and certainly discussion.
- Discussion needs to be shortened focusing on the possible implications of the affected genes, the differences with other reports and so on.
Minor comments
- Key words are missing
- Reference 1 is not the correct one for the sentence, same for ref 4. I recommend being careful about the citations used in the manuscript, please revise all references along the manuscript
- Line 54, rewriting: change “undergo” by “produce”
- Lines 93-100, this is part of the methods section, please move
- Lines 124-125, please indicate the frequency of such HPV genotypes detected.
- Lines 127-131 need an appropriated reference
Author Response
Reviewer 2
Comments and Suggestions for Authors
The manuscript presented by Garza-Rodriguez et al., describes the potential HPV integration sites in different cervical lesions. Certainly, the authors provide new insights for understanding the biological manners that could be potentially associated to cancer development. The manuscript is well written, and the methodology is very innovative, providing new approaches for expanding our knowledge in the field. Nevertheless, several important aspects are missing, and the manuscript must be improved before to be considered for publication.
Major concerns
- Complete information of tables 1 and 2 could go to supplementary information, these only show HPV detected in the studies population. Instead, authors could present this information as a new table indicating the frequencies of the HPV in the included analyzed cases to increase readability.
Answer: Our central idea is about diversity of HPV integrations. Tables 1 and 2 focus in showing that samples may contain multiple HPV types (co-infection). Therefore, we believe the information of Table 1 & 2 are important in main paper to interpret the following results of the diverse HPV-type integrations. We appreciate the opportunity to clarify this.
- Authors indicate that they distinguished variants in their samples, but they do not present such information, this information can be presented in a table due to the available information showing that variants confer a different prognosis in CC.
Answer: The variants in the discussion were mentioned to distinguish samples and prove that readings are not a product of possible contamination. What we identified are nucleotide variant sites. To clarify this, we change the word “variants” by “variant sites” and in some paragraphs was changed by “nucleotide variant sites” (Lines 149, 155, 175, 177, 297, and 298). We also added a reference to the figure 2 showing the referred variant sites in the discussion. We appreciate the comment helping to clarify this issue.
- Why the authors included the case with an unknown result of PAP, since they pretend to associate the integration with the different lesions. This must be clarified.
Answer: Although the PAP smear was difficult to interpret in 2 cases from the transcripts, all samples were HPV+ before the study and added the reference to our previous analysis. We added this in the section of “Study Population” in methods. We appreciate you spotted this omission. The paragraph includes (Lines 486-487):
“The PAP smear result was re-interpreted from transcripts in all but two samples. All the patients included were previously detected as HPV positive [63].”
- It is not clear whether the integrations spots found in the study are associated to the different clinical stages analyzed in the study. Thus, authors need to present such information in an additional figure, since this is basically the main goal of this work.
Answer: Our analysis includes 13 LSIL, 2 HSIL, 3 Normal, 2 ASCUS, and 1 CC. Thus, it was not possible to analyze integrations across clinical pap smear subtypes. Nevertheless, we added the clinical pap smear result in all samples of Figure 5 to include clinical information with integrations and associate them visually.
- Authors indicate that these results could be associated to clinical stage, viral persistence and other clinical manifestations (lines 76-79) but is not covered in the manuscript. This needs clarification and certainly discussion.
Answer: In section “3.2 HPV Integration” we discuss that integrations were detected in “apparently” normal samples and early pre-cancerous samples. We added a mention of viral persistence since we have not studied their integrations yet.
- Discussion needs to be shortened focusing on the possible implications of the affected genes, the differences with other reports and so on.
Answer: We dedicated a discussion section for human genes “3.6. Comparison of Reported Human Genes”. We showed that known integrations in particular genes were found. Moreover, we also found hundreds of novel integrations. We mention that 2616 genes show at least two integrations sites. Thus, it is difficult to choose novel specific genes. We attempted to focus on repetitive genes such as those shown in figures 5 and 6.
Minor comments
- Key words are missing
Answer: We already added the keywords as follows (Line 41):
“Keywords: Cervical cancer, HPV integration, hot spots, cervical lesions.”
- Reference 1 is not the correct one for the sentence, same for ref 4. I recommend being careful about the citations used in the manuscript, please revise all references along the manuscript.
Answer: We revised all the references.
Reference 1 was changed by:
Wold Health Organization / Cervical cancer: Overview. https://www.who.int/health-topics/cervical-cancer#tab=tab_1 (February 9, 2021).
Reference 4 was deleted from the line 46. The rest of the references were revised.
- Line 54, rewriting: change “undergo” by “produce”
Answer: We changed the word “undergo” by “produce” (Lines 57 and 59).
- Lines 93-100, this is part of the methods section, please move.
Answer: These lines are meant to summarize our approach to facilitate reading in the format of this journal where methods are shown at the end of the manuscript.
- Lines 124-125, please indicate the frequency of such HPV genotypes detected.
Answer: The frequency of HPV types can be inferred from Table 2. Because many of the patients have multiple HPV infections, we believe that the frequency table could confuse the reader.
- Lines 127-131 need an appropriated reference.
Answer: To facilitate the estimation of depth, we added data for calculations like 140,000 * 120 / 8000).
Round 2
Reviewer 2 Report
Authors attended all suggestions. The manuscript is now suitable for publication.